# Rho/ROCK Pathway and Noncoding RNAs: Implications in Ischemic Stroke and Spinal Cord Injury

**DOI:** 10.3390/ijms222111573

**Published:** 2021-10-26

**Authors:** Tetsu Kimura, Yuta Horikoshi, Chika Kuriyagawa, Yukitoshi Niiyama

**Affiliations:** Department of Anesthesia and Intensive Care Medicine, Akita University Graduate School of Medicine, Akita 010-8543, Japan; y.horikoshi@med.akita-u.ac.jp (Y.H.); chibi_hamaguri315@yahoo.co.jp (C.K.); niiyama@med.akita-u.ac.jp (Y.N.)

**Keywords:** stroke, spinal cord injury, Rho, Rho kinase, noncoding RNA, apoptosis, inflammation, axon regrowth, neurogenesis, angiogenesis

## Abstract

Ischemic strokes (IS) and spinal cord injuries (SCI) are major causes of disability. RhoA is a small GTPase protein that activates a downstream effector, ROCK. The up-regulation of the RhoA/ROCK pathway contributes to neuronal apoptosis, neuroinflammation, blood-brain barrier dysfunction, astrogliosis, and axon growth inhibition in IS and SCI. Noncoding RNAs (ncRNAs), such as microRNAs (miRNAs) and long noncoding RNAs (lncRNAs), were previously considered to be non-functional. However, they have attracted much attention because they play an essential role in regulating gene expression in physiological and pathological conditions. There is growing evidence that ROCK inhibitors, such as fasudil and VX-210, can reduce injury in IS and SCI in animal models and clinical trials. Recently, it has been reported that miRNAs are decreased in IS and SCI, while lncRNAs are increased. Inhibiting the Rho/ROCK pathway with miRNAs alleviates apoptosis, neuroinflammation, oxidative stress, and axon growth inhibition in IS and SCI. Further studies are required to explore the significance of ncRNAs in IS and SCI and to establish new strategies for preventing and treating these devastating diseases.

## 1. Introduction

Ischemic strokes (IS) and spinal cord injuries (SCI) are major causes of disability worldwide. Patients affected by these diseases require long-term care and lose social productivity. More importantly, the emotional burden on patients and their families is immeasurable. The sudden cessation of blood supply or mechanical insult triggers the secondary injury cascade that induces further permanent damage. Although more than 200 neuroprotective agents have been developed and evaluated in animal and clinical trials, few have been applied clinically [1]. In IS and SCI, neuronal apoptosis, inflammation, oxidative stress, and excitotoxicity have been identified as secondary injury mechanisms [2]. However, in recent years, the involvement of non-neuronal cells, such as astrocytes, vascular endothelial cells, and microglia, has been attracting attention and is being studied as a therapeutic target [3].

RhoA is a small GTPase protein that belongs to the Rho GTPase family, including Rho, Rac, Cdc42, Rnd, RhoD, RhoBTB, and RhoH. Rho-associated coiled-coil protein kinase (ROCK) is a downstream effector of RhoA. The Rho/ROCK pathway regulates a variety of critical cellular functions such as gene transcription, cell-cell adhesion, cell cycle progression, dendritic arborization, spine morphogenesis, growth cone development, axon guidance, neuronal survival, and neuronal death [4]. Since excessive Rho/ROCK activity contributes to the pathophysiology of a wide range of disorders, such as subarachnoid hemorrhage, retinal disease, epilepsy, Parkinson’s disease, Alzheimer’s disease, IS, and SCI, many researchers have pursued the potential of this pathway as a therapeutic target [5]. Accumulating evidence suggests that inhibition of the Rho/ROCK pathway may be effective in treating these diseases [6,7].

Micro RNAs (miRNAs) and long noncoding RNAs (lncRNAs) are members of the noncoding RNA (ncRNA) family and were previously considered to have no function. MiRNAs and lncRNAs have gained much attention in recent years because they play essential roles in many biological functions and are also deeply involved in various pathological conditions, including ischemia-reperfusion injuries [8].

This review outlines the significance of the Rho/ROCK pathway in IS and SCI and the involvement of ncRNAs in the Rho/ROCK pathway in their pathophysiology, which has been reported recently.

## 2. Rho/ROCK Pathway

Rho is inactive when bound to GDP while becoming active when bound to GTP via guanine exchange factors (GEP). GTP-bound activated Rho activates its downstream effector, ROCK. ROCK is a serine/threonine kinases family and includes two isoforms: ROCK1 and ROCK2 [9]. ROCK 1 transcript is prominently expressed in non-neuronal tissues, while ROCK 2 is present more abundantly in the brain and skeletal muscle [9]. Activated ROCK then phosphorylates multiple downstream effectors, including myosin light chain (MLC), myosin light chain phosphatase (MLCP), LIM kinase, ezrin/radixin/moesin (ERM), collapsin response mediator protein 2 (CRMP2), adducin, and so on. As a result, ROCK regulates smooth muscle contraction, cytoskeletal rearrangement via stress fiber formation, focal adhesion, actin filament stabilization, growth cone collapse, and actin network assembly (Figure 1) [10,11,12]. ROCK is also involved in apoptosis via the cleavage of caspase-3 or granzyme B [12]. Furthermore, the inhibition of ROCK significantly reduced focal adhesion and stress fiber formation induced by Thy-1 (CD90) in astrocytes, suggesting the importance of the Rho/ROCK pathway in the processes involved in neuron-glia communication in the brain [13]. The Rho/ROCK pathway contributes to many pathological conditions such as cardiovascular diseases, cancer, neurological diseases, Alzheimer’s disease, IS, and SCI [14,15,16].

## 3. Noncoding RNAs

MiRNAs are endogenously expressed small-noncoding RNAs consisting of 20–22 nucleotides that regulate gene expression at a posttranscriptional level through their interaction with the 3′-untranslated region (UTR) of the target messenger RNAs (mRNAs) [17]. MiRNAs are first transcribed from genomic DNA and then transcribed into primary RNA (pri-RNA) by RNA polymerase II. The pri-RNA is several thousand base pairs long and consists of at least one hairpin loop. This hairpin loop is recognized and cleaved by the endonuclease Drosha to generate precursor miRNAs (pre-miRNAs) with the help of the double-stranded RNA-binding protein DiGeorge syndrome critical region 8 (DGCR8) [18,19,20]. Pre-miRNAs are transported from the nucleus to the cytoplasm by the intervention of exportin-5. Pre-miRNAs are cleaved by endoribonuclease Dicer to form a duplex of biologically active mature miRNA strands in the cytoplasm [21,22,23]. MiRNAs regulate various cellular functions, including neuronal development, differentiation, synaptic plasticity, proliferation, and metabolism [24]. Previous studies have shown that miRNAs are deeply involved in stroke pathology through oxidative stress, neuroinflammation, apoptosis, and vascular endothelial damage [25,26].

LncRNAs are a family of ncRNAs comprising more than 200 nucleotides that regulate gene expression via various mechanisms [27]. Recent studies revealed that lncRNAs act as competing endogenous RNAs (ceRNAs) that sponge specific miRNAs to regulate gene expressions.

## 4. Pathophysiology of IS

Strokes are a significant cause of disability and mortality worldwide, resulting in more than 6 million deaths each year [28]. It is estimated that 80% of strokes are ischemic strokes, 15% are hemorrhagic strokes, and the remaining 5% have unknown causes [29]. IS occurs by the cessation of the blood supply to the brain. In an IS, the oxygen and energy supply to the neurons is deprived, causing them to stop functioning after a few seconds and undergo structural changes after only 2 min [30]. Following ischemia, the depletion of glucose and oxygen to the neurons causes the lack of adenosine triphosphate (ATP), then ion pump failure occurs. The imbalance of ion concentrations inside and outside the cells causes cytotoxic edema, releasing excitatory neurotransmitters such as glutamate and aspartate [31,32]. Oxygen depletion leads to anaerobic metabolism, resulting in metabolic acidosis. All these events contribute to necrosis. While severe ischemia occurs in the ischemic core, causing neuronal cell necrosis, the surrounding penumbra region cells are partially injured, with the potential to be salvaged [33]. However, cerebral ischemia induces an ischemic cascade that also causes neuronal death in the penumbra region.

Ischemic insults also trigger stress signals and the upregulation of immediate early genes, causing mitochondrial dysfunction and apoptosis [34]. Furthermore, reactive oxygen species produced by reperfusion leads to vascular endothelial damage, disrupting the blood-brain-barrier (BBB) via activating matrix metalloproteinases (MMPs) [35]. Ischemic insult further triggers the upregulation of pro-inflammatory mediators, which induces the expression of adhesion molecules, leading to neutrophil recruitment, attachment, and transmigration from the blood into the brain parenchyma followed by macrophages and monocytes. Toxic mediators produced by activated inflammatory cells and injured neurons, such as cytokines, nitric oxide, superoxide anion, and prostanoids, deteriorate tissue injuries [36].

On the other hand, ischemia triggers phosphate protein kinase B (AKT) activation and the upregulation of trophic factors, leading to recovery and repair mechanisms, including angiogenesis, neurogenesis, and synaptogenesis [37].

## 5. Rho/ROCK Pathway in IS

The up-regulation of the Rho/ROCK pathway in neurons and astrocytes after a stroke has been reported. In rodent models of middle cerebral artery occlusion (MCAO), ROCK activity in the ischemic area increased. The administration of Fasudil, a ROCK inhibitor, suppressed the ROCK activation, increased cerebral blood flow, reduced infarct size, and improved neurologic outcomes [38,39]. Fasudil administration 6 and 24 h after ischemia also inhibited neuronal cell death and reduced infarct size, indicating a broad therapeutic time window [40,41]. As described below, the Rho/ROCK pathway plays a crucial role in strokes, including apoptosis, excitotoxicity, platelet function, neuroinflammation, blood-brain barrier (BBB), astrocytes, axon growth inhibition, and neurogenesis/angiogenesis.

### 5.1. Rho/ROCK Pathway and Neuroprotection in IS

RhoA induced cell death in the rodent stroke model by activating ROCK, which phosphorylates phosphatase and tensin homolog deleted from chromosome 10 (PTEN) and inactivates AKT [42]. Further, ROCK inhibition with fasudil prevented ischemia-induced neuronal apoptosis by maintaining the AKT signaling pathway [42]. Recently, Wang et al. found that non-muscle myosin heavy chain (NMMHC) IIA inhibition attenuated neuronal apoptosis, and this effect was related to the caspase-3/ROCK/myosin light chains (MLC) signaling pathway [43]. ROCK inhibitors, fasudil or Y-27632, also prevented cell death due to excitotoxicity [44].

### 5.2. Rho/ROCK Pathway and Platelet Function in IS

ROCK2 deficient platelets were less responsive to thrombin stimulation, such as pseudopodia formation, collagen adhesion, and heterotypic aggregation, leading to prolonged bleeding time and an increase in time to vascular occlusion [45]. Therefore, ROCK2 is essential for forming and stabilizing blood clots and may be an essential mediator of thromboembolic strokes [45].

### 5.3. Rho/ROCK Pathway and Neuroinflammation in IS

Although inflammatory responses after ischemic insult help isolate the injured region, excessive inflammation may deteriorate ischemic injury [46].

ROCK activation after a stroke contributes to the deterioration of cerebral injury in the acute phase by stimulating neuroinflammation. ROCK mediates the overexpression of adhesion molecules, such as P-selectin and intercellular adhesion molecule (ICAM)-1, via endothelium-derived nitric oxide synthase (eNOS) reduction [47]. On the other hand, ROCK inhibition reduced neutrophil accumulation in the ischemic region and reduced the infarct volume [48,49]. ROCK activation in microglia, a resident macrophage in the brain, leads to pro-inflammatory cytokine secretion [50]. ROCK inhibition with fasudil reduced hippocampal injury by suppressing the microglial secretion of pro-inflammatory cytokines [51].

### 5.4. Rho/ROCK Pathway and BBB in IS

BBB comprises endothelial cells, pericytes, perivascular antigen-presenting cells, astrocytic endfeet, and parenchymal basement membrane [52].

The neurovascular unit (NVU) is a conceptual model that emphasizes the dynamic interactions between neurons and components of the BBB, such as astrocytes, smooth muscle cells, endothelial cells, pericytes, and basement membranes, as well as supporting cells, microglia, and oligodendroglia, which are necessary for normal brain function [53]. ROCK activation after a stroke promoted microvascular damage by upregulating MMP-9 [54]. Monocyte chemoattractant protein-1 (MCP-1) altered actin and tight junction structure reorganization and, thus, permeability through RhoA/ROCK activation [55]. A-kinase anchor protein 12 (AKAP12) alleviated the damage and dysfunction of the BBB after ischemia through the suppression of ROCK [56]. Thus, ischemia-induced activation of the Rho/ROCK pathway disrupts the BBB, leading to NVU dysfunction.

### 5.5. Rho/ROCK Pathway and Astrocytes in IS

Astrocytes play an essential role in energy storage and transfer to keep normal neurotransmission, neurotransmitter reuptake and recycling, and the maintenance of ion homeostasis [52]. Specific interaction between endothelial cells and astrocytes of NVU is essential to regulate BBB under normal and pathological conditions [57]. Under IS, several pathological events, including fibrin accumulation, the transmigration of leukocytes, the production of degrading enzymes, and basal laminae breakdown with loss of astrocytes and endothelial cells, contribute to the breakdown of BBB, resulting in vasogenic edema and hemorrhagic transformation [58].

The Rho/ROCK pathway becomes upregulated in astrocytes after the stroke [59,60]. Over-activated astrocytes, called reactive astrocytes, change their morphology and retract their endfeet connections from the blood vessels and neurons via Rho/ROCK pathway activation, leading to the breakdown of NVU coupling and scar formation, which is called reactive astrogliosis [52,61].

### 5.6. Rho/ROCK Pathway and Axon Growth Inhibition in IS

Rho/ROCK pathways are involved in the pathophysiology of various diseases, including IS, SCI, optic nerve injury, and inflammatory diseases. Although the axons in the peripheral nervous system show the capacity to regenerate after injury, CNS axons show limited regrowth capacity. When myelin, ensheathing axons and composed of oligodendrocytes, is injured, the CNS axons are exposed to myelin debris that contains myelin-associated inhibitors (MAI). MAI, such as Nogo, myelin-associated glycoprotein (MAG), and oligodendrocyte-myelin glycoprotein (OMgp), are expressed in the oligodendrocytes and transduce signals to neurons through the Nogo receptor (NgR), leading to the Rho/ROCK pathway activation and resulting in axon growth inhibition [62].

Repulsive guidance molecule (RGM) is a protein that induces growth cone collapse and has three homologs, including RGMa, RGMb, and RGMc [63]. RGMa expression is increased after SCI and inhibits axon regeneration [64]. Binding RGMa to its receptor neogenin activates the RhoA/ROCK pathway, leading to neurite outgrowth inhibition [62]. Neutralizing anti-RGMa antibodies promoted axonal regeneration and functional recovery after SCI in rats [65].

Reactive astrocytes, forming a glial scar, secrete inhibitory extracellular matrix molecules at the lesion site, including chondroitin sulfate proteoglycans (CSPGs). CSPGs activate the RhoA/ROCK pathway through protein tyrosine phosphatase (PTPσ), NgR, and leukocyte common antigen-related phosphatase (LAR), resulting in axon growth inhibition [66].

### 5.7. Rho/ROCK Pathway and Neurogenesis/Angiogenesis in IS

Neurogenesis is defined as the production of new functional neurons from neural stem cells (NSCs) and comprises the proliferation of NSCs, migration, and differentiation into mature neurons. Although neurogenesis occurs in the normal brain, IS also triggers enhanced neurogenesis [67]. Angiogenesis, the new microvessel formation through the ramification from pre-existing blood vessels, comprises endothelial cell proliferation and sprouting, forming tube-like structures, branching, and anastomosis. Angiogenesis occurs in the penumbra region after IS [68]. Furthermore, recent studies revealed that the interaction between angiogenesis and neurogenesis is crucial to enhance brain reparation after IS [69]. For instance, endothelial cells activated by ischemia secrete the regulatory factors to modulate NSCs, leading to neurogenesis [70], while NSCs also secrete several factors to promote angiogenesis [71].

The ROCK inhibitor fasudil upregulated astrocytes to produce the granulocyte colony-stimulating factor (G-CSF), leading to inducing neurogenesis and neuroprotection under oxygen-glucose deprivation (OGD) [72]. Therefore, the Rho/ROCK pathway may block the neurogenesis, resulting in worsening neuronal recovery.

Sonic hedgehog (Shh), a soluble protein that upregulates angiogenic growth factors, is secreted from astrocytes under oxidative stress [73]. Because inhibiting the RhoA/ROCK pathway diminished the angiogenesis induced by astrocyte-derived Shh after OGD [74], the RhoA/ROCK pathway may be involved in astrocyte-mediated angiogenesis [75]. However, constraint-induced movement therapy and fasudil promoted angiogenesis and neurogenesis after cerebral ischemia by overcoming the Nogo-A/Rho/ROCK pathway [76]. Further studies are needed to elucidate whether inhibiting Rho/ROCK pathway promotes angiogenesis or not.

## 6. Rho/ROCK Pathway in SCI

As in strokes, myelin-associated molecules, such as Nogo, MAG, OMgp, semaphorin 4D, ephrin B3, RGM, and netrin-I and glial scar-associated extracellular matrix molecules known as CSPGs, converge on the Rho/ROCK pathway, resulting in the inhibition of axon regeneration [62]. Indeed, Rho inhibition contributes to axon regrowth and neuroprotection after a spinal cord injury [77,78,79]. VX-210 (cethrin, BA-210), which deactivates RhoA, improved neurological outcomes in patients with SCI [80].

Fasudil was also neuroprotective after SCI and spinal cord ischemia in a rat model in vivo [81,82,83]. However, upregulating the cyclooxygenase (COX)-2 pathway caused resistance against ROCK inhibitors [84]. Kim et al. examined the effects of combined treatment with fasudil and menthol, a natural compound, which reduces glutaminergic neurotoxicity, decreases inflammation, and suppresses COX-2 expression [85]. They found that combined treatment of fasudil and menthol improved the functional recovery after SCI by alleviating apoptosis, inflammation, and glial scar formation and promoting neovascularization.

Microglia has a crucial role in the immune system of the central nervous system, which responds in a few minutes and converges to the damaged site after injury [86]. Microglia released pro-inflammatory cytokines, which exacerbate tissue damage while phagocytosing tissue debris and pathogens to mitigate damage [87]. After SCI, RhoA activation also occurs in microglia [88]. Because ROCK inhibitors, fasudil and Y27632, promoted microglial migration and initiated cell morphological changes through the extracellular signal-regulated kinase (ERK) signaling pathway [89], the Rho/ROCK pathway may be involved in the microglial migration after SCI. In addition, the ERK pathway played a crucial role in the Y27632- and fasudil-induced changes in microglial morphology. Rho guanine nucleotide exchange factor 3 (Arhgef3) is part of the Rho guanine nucleotide exchange factors (RhoGEFs) family, with high selectivity to RhoA and RhoB [90]. Disrupting Arfgef3 expression attenuated microglial inflammation and protected neuronal tissues from secondary damage after SCI via the inhibition of RhoA activation [91].

In rodent models, RhoA-inhibitors, β-elemene, Leucine-rich repeats and Ig domain-containing Nogo receptor interacting protein-1 (LINGO-1)-Fc, ibuprofen, small interfering RhoA (siRhoA), RhoA+FK506, fasudil, p21Clip1/WAF1, Y27632, and VX-210 have neuroprotective properties after SCI, including axon sprouting, regenerating nerve fibers, reducing the formation of syrinx-cavity, and protecting white matter, leading to the recovery of locomotor function [92].

## 7. ncRNAs in IS

Several studies have demonstrated the relationship between miRNAs and the pathogenesis of ischemic stroke, including excitotoxicity, neuroinflammation, and neuronal death. Furthermore, there is growing evidence that miRNAs are associated with angiogenesis, neurogenesis, and neuroprotection after IS [93]. Although the target genes of miRNAs associated with IS are diverse, miRNAs related to the Rho/ROCK pathway have only been reported in the last few years.

## 8. Rho/ROCK Pathway and ncRNAs in IS

### 8.1. Rho/ROCK Pathway and miRNAs for Apoptosis in IS

MiR-190 was downregulated after cerebral injury [94]. Using the MCAO-reperfusion (MCAO/R) model, Jiang et al. showed that Rho was a direct target of miR-190 and that the overexpression of miR-190 reduced brain damage and apoptosis via the Rho/ROCK pathway [95] (Table 1 and Figure 2).

Han et al. investigated the neuroprotective effects of miR-431 using the rat MCAO/R model [96]. In their model, the expression of miR-431 significantly reduced while that of Rho significantly increased. Rho was the potential target gene of miR-431, and miR-431 negatively regulated Rho expression in hippocampal neurons. They concluded that miR-431 promoted proliferation and inhibited apoptosis by negatively regulating the Rho/ROCK pathway (Table 1 and Figure 2).

The anti-apoptotic effect of miR-335 and the correlation between stress granules (SG) formation and apoptosis in acute IS was investigated by Si et al. [97]. SGs are complex and dynamic foci generated in the cytoplasm when eukaryotic cells suffer from different types of stress, e.g., endoplasmic reticulum stress, heat shock, and acute energy starvation [98]. In the eukaryotes under stress, multiple proteins called RNA binding proteins (RBPs), including T-cell intracellular antigen-1 (TIA1), self-aggregate to form the SG. SG formation protects mRNA and proteins against degradation and misfolding, enhancing cellular resistance to apoptosis [99]. They found that SG formation promoted by miR-335 suppressed apoptosis by inhibiting the expression of ROCK2 (Table 1 and Figure 2).

Ding et al. explored the role of miR-582-5p and proteinase-activated receptors type-1 (PAR-1) after ischemia-reperfusion [100]. PAR-1 is a thrombin receptor, and its deficiency protects against neuronal damage after ischemia-reperfusion [101,102]. They found that miR-582-5p expression decreased, and PAR-1, RhoA, and ROCK2 increased, after ischemia-reperfusion. The overexpression of miR-582 reduced neuronal apoptosis by inhibiting the Rho/ROCK pathway through the downregulation of PAR-1 (Table 1 and Figure 2).

### 8.2. Rho/ROCK Pathway and lncRNAs/miRNAs for Apoptosis in IS

LncRNAs may be associated with cell apoptosis, inflammation, cell death, and angiogenesis in IS [103,104,105,106]. Therefore, lncRNAs have been emerging as a new therapeutic target in IS [107]. One of these lncRNAs is the X-inactive specific transcript (XIST) RNA, a 17-kb lncRNA, which regulates X chromosome inactivation in mammals. There is a report of the up-regulation of XIST enhanced cerebral ischemia-reperfusion injury in SH-SY5Y cells [108]. Wang et al. investigated the relationship among XIST, miR-362, and ROCK2 in an MCAO/R model [109]. They found that XIST negatively regulated miR-362, and the depletion of XIST attenuated ischemia-reperfusion induced apoptosis and inflammatory responses by regulating the miR-362/ROCK2 axis (Table 1 and Figure 2).

### 8.3. Rho/ROCK Pathway and lncRNAs/miRNAs for Oxidative Stress/Inflammation in IS

Zeng et al. investigated whether metformin, a commonly used drug for treating type 2 diabetes, prevents cerebral ischemic injury through its antioxidant effect via the modulation of the lncRNA-H19/miRNA-148a-3p/ROCK2 pathway [110]. Elevated expression of lncRNA-H19 related to the progression of cerebral ischemia [111]. They found that metformin protected against cerebral damage via inhibiting oxidative stress and apoptosis. In addition, the expression of lncRNA- H19 and ROCK2 increased, and the miR-148a-3p expression decreased after ischemia-reperfusion, while metformin inhibited these responses. Thus, they concluded that metformin exerted neuroprotective effects against ischemia-induced cerebral injury by inhibiting oxidative stress and apoptosis by regulating the lncRNA-H19/miR148a-3p/ROCK2 axis (Table 1 and Figure 2).

Using the MCAO/R model and OGD/R treated PC12 cells, Zhong et al. revealed the upregulation of lncRNA small nucleolar RNA host gene 14 (SNHG14) [112]. LncRNA-SNHG14 negatively regulated miR-136-5p as its competing endogenous RNA (ceRNA). Inhibiting SNHG14 decreased neuronal injury and inflammation, and SNHG14 positively regulated the expression of ROCK1 by acting as a sponge of miR-136-5p. Therefore, SNHG14 silencing improved neurological function and prevented inflammation dependent on miRNA-136-5p overexpression and decreased ROCK1 level [112] (Table 1 and Figure 2).

Chen et al. investigated the expression of the lncRNA regulator of reprogramming (ROR) in cerebral hypoxia-reoxygenation in PC12 cells and analyzed the effect of lncRNA-ROR on the ROCK1/ROCK2 signaling pathway [113]. lncRNA-ROR promoted human-induced pluripotent stem cells and participated in miRNA-mediated suppression in human embryonic stem cell self-renewal [114]. They found that miR135a-5p was a direct target gene of lncRNA-ROR. The overexpression of lncRNA-ROR induced by hypoxia-reoxygenation aggravated the oxidative damage and apoptosis of PC12 cells by inhibiting the expression of miR135a-5p. Furthermore, miR135a-5p overexpression decreased the damage by inhibiting the expression of ROCK1/2 (Table 1 and Figure 2).

## 9. Rho/ROCK Pathway and miRNAs for Apoptosis/Axon Regeneration in SCI

Recently, increasing studies have focused on regulating miRNAs in promoting axon outgrowth and inhibiting neuronal apoptosis [115,116,117]. MiRNAs bind to the target messenger RNAs (mRNAs) and negatively regulate gene expression at both the mRNA and protein levels [118] (Table 2 and Figure 3).

Semaphorin-3A (Sema3A) is a neuronal secreted repulsive guidance cue and induces neuronal growth cone collapse during the development of the nervous system. Sema3A binds to the receptor complex containing PlexinA1 and Neuropilin-1 (NRP-1) and modulates the Rho/ROCK pathway [119]. Wang et al. investigated whether miR-30b, which targets sema3A to promote retinal ganglion cell neurite growth [120], could exert primary sensory neuron neurite outgrowth after SCI [121]. They found that the up-regulation of miR-30b inhibited sema3A expression and RhoA/ROCK activity through the PlexinA1/NRP-1 co-receptor, promoting primary sensory neuron neurite outgrowth and spinal cord sensory conductive function recovery (Table 2 and Figure 3). Interestingly, the reduced expression of miR-30b has been proposed as one of the biomarkers for ischemic strokes [122], suggesting that miRNAs are not only promising targets for the treatment of ischemic strokes but are also valuable as biomarkers for diagnosis.

Extracellular vesicles (EVs) are candidates for the vehicles of bioactive molecules, such as mRNA, miRNAs, and lncRNAs, and EVs derived from various cells are thus expected to be a potential therapeutic method. Jia et al. investigated whether miR-381 encapsulated in the EVs derived from mesenchymal stem cells (MSCs) can promote the recovery of SCI [123]. MiR-138 is essential for the proliferation of nerve cells during SCI. Furthermore, it may be associated with bromodomain-containing protein-4 (BRD4), which can bind to Wnt family member 5A (WNT5A). WNT5A reportedly inhibits axon growth and stimulates cell apoptosis [124,125]. They found that miR-381 delivered by EVs derived from MSCs inhibited neuron apoptosis and promoted the recovery of SCI by inhibiting the BRD4/WNT5A axis (Table 2 and Figure 3).

MiR-135a-5p stimulated axon regrowth and inhibited apoptosis [126,127]. Therefore, Wang chose specificity protein 1 (SP1) and Rho-associated kinase (ROCK) as target genes of miR-135a-5p because these proteins were known to regulate neural apoptosis and axon regeneration [128,129,130]. SP1 binds to genes associated with apoptosis, such as Bax, Bcl-2, and caspase 3, and activates the apoptosis pathway [131,132,133]. Meanwhile, ROCK is considered a direct target gene of miR-135a-5p [134,135]. The AKT/glycogen synthase kinase 3β (GSK3β) pathway is one of the downstream signaling pathways of ROCK, and the activation of this pathway regulates axonal growth [136,137]. They found that the miR-135a-5p-SP1 axis regulated the Bax/Bcl-2/caspase-3 signaling pathway to modulate neuronal apoptosis. The miR-135a-5p-ROCK axis regulated the AKT/GSK3β signaling pathway to promote axon regeneration during the process of functional recovery following SCI (Table 2 and Figure 3). They proposed that the genetic manipulation of cells according to these two signaling axes may be candidates for the clinical translation of stem cell therapy.

**Table 2 ijms-22-11573-t002:** MiRNAs and the Rho/ROCK pathway in SCI.

Model	miRNA	Expression after Insult	Target	Effects	Reference
SDCL in rat Primary sensory neuron of rat	miR-30b	Decreased	Sema3A	miR-30b agomir promoted neurite outgrowth, and antagomir inhibited it. miR-30b agomir regulates sema3A/PlexinA1-NRP-1/RhoA/ROCK pathway, promoting sensory conductive function recovery after SDCL.	[121]
SCI in rat DRG cells of rat	miR-381	Decreased	BRD4	BRD4 promoted WNT5A expression via binding to the promotor of WNT5A. WNT5A promoted apoptosis by activating the RhoA/ROCK pathway. miR-381 derived from EV in MSCs inhibited neuron apoptosis and promoted the recovery of SCI by inhibiting the BRD4/WNT5A axis.	[123]
SCI in rat PC12 cells under H_2_O_2_ stimulation	miR-135a-5p	Decreased	SP1 ROCK1/2	miR-135a-5p-SP1-Bax/Bcl-2/caspase3 axis inhibited neuronal apoptosis. miR-135a-5p-ROCK-AKT/GSK3β pathway promoted axon regeneration during functional recovery after SCI.	[130]

SCI: Spinal cord injury; Sema3A: Semaphorin-3A; SDCL: Spinal cord dorsal column lesion; DRG: Dorsal root ganglia; BRD4: Bromodomain-containing protein 4; WNT5A: Wnt family member 5A; EV: extracellular vesicles; MSCs: mesenchymal stem cells; SP1: Specificity protein 1; AKT: phosphate protein kinase B; GSK3: glycogen synthase kinase 3β.

## 10. Conclusions

As we have discussed, the Rho/ROCK pathway is deeply involved in IS and SCI in various ways, including apoptosis, neuroinflammation, BBB integrity, astrogliosis, axonal regeneration, neurogenesis, and angiogenesis. Furthermore, there is growing evidence that the inhibition of the Rho/ROCK pathway can effectively reduce IS and SCI-induced injury. Here, we introduced the involvement of lncRNAs and miRNAs through the Rho/ROCK pathway in IS and SCI and the potential therapeutic effects of the intervention with these ncRNAs. The clinical utility of miRNAs will be very high if they can be delivered in the form of extracellular vesicles, as shown by Jia et al. [123]. Because previous studies on the relationship between the Rho/ROCK pathway and ncRNAs in IS and SCI were only related to apoptosis, neuroinflammation, oxidative stress, and axonal regeneration, future studies are expected to include BBB integrity, astrogliosis, neurogenesis, and angiogenesis.

In this review, we focused on the potential of the Rho/ROCK pathway and ncRNAs as therapeutic targets in IS and SCI. However, it has been suggested that leukocyte ROCK activity is an independent predictor of cardiovascular morbidity and mortality, including strokes [138]. Furthermore, as mentioned earlier, circulating miRNAs, including miR-30b, have been proposed as valuable biomarkers for diagnosing IS [122]. Therefore, ncRNAs targeting the Rho/ROCK pathway may be worthy of further investigation not only as therapeutic targets for IS and SCI but also as novel biomarkers.

In conclusion, the Rho/ROCK pathway is now one of the most attractive targets for treating IS and SCI because of its deep involvement in a wide range of pathological conditions. The inhibition of the Rho/ROCK pathway using ncRNAs is a promising therapeutic approach that warrants further investigation.

## Figures and Tables

**Figure 1 ijms-22-11573-f001:**
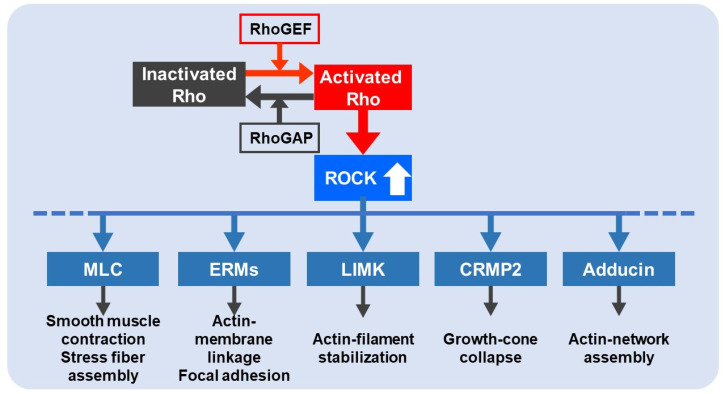
Representative downstream effectors of the Rho/ROCK pathway involved in fundamental biological roles in cellular function. RhoGEF: Rho guanine nucleotide exchange factor; RhoGAP: Rho GTPase activating protein; MLC: myosin light chain; ERM: ezrin/radixin/moesin; LIMK: LIM kinase; CRMP2: collapsin response mediator protein 2.

**Figure 2 ijms-22-11573-f002:**
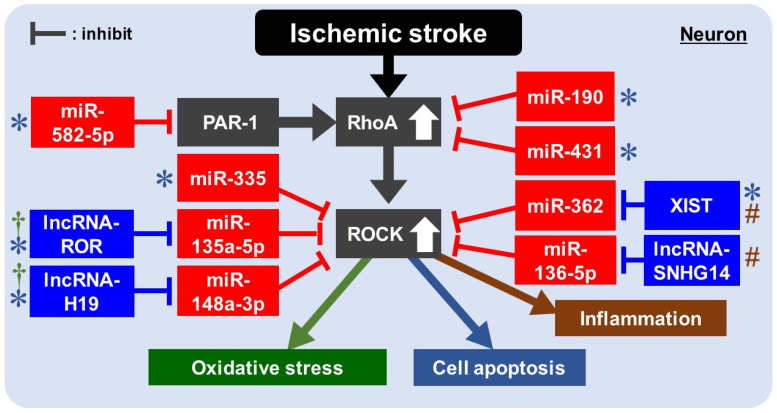
Schematic representation of the interaction between the Rho/ROCK pathway and ncRNAs in ischemic strokes. Ischemic strokes trigger multiple detrimental pathways, including the RhoA/ROCK pathway, leading to oxidative stress, cell apoptosis, inflammation, and excitotoxicity. MiRNAs inhibit oxidative stress, cell apoptosis, and inflammation via the inhibition of RhoA or ROCK upregulation in neurons. LncRNAs exacerbate ischemic injuries via the downregulation of miRNAs. † miRNA/lncRNA involving oxidative stress. * miRNA/lncRNA involving cell apoptosis. # miRNA/lncRNA involving inflammation. PAR-1: proteinase-activated receptor type I; ROR: regulator of reprogramming; XIST: X-inactive specific transcript; SNHG14: small nucleolar RNA host gene 14; ROCK: Rho-kinase.

**Figure 3 ijms-22-11573-f003:**
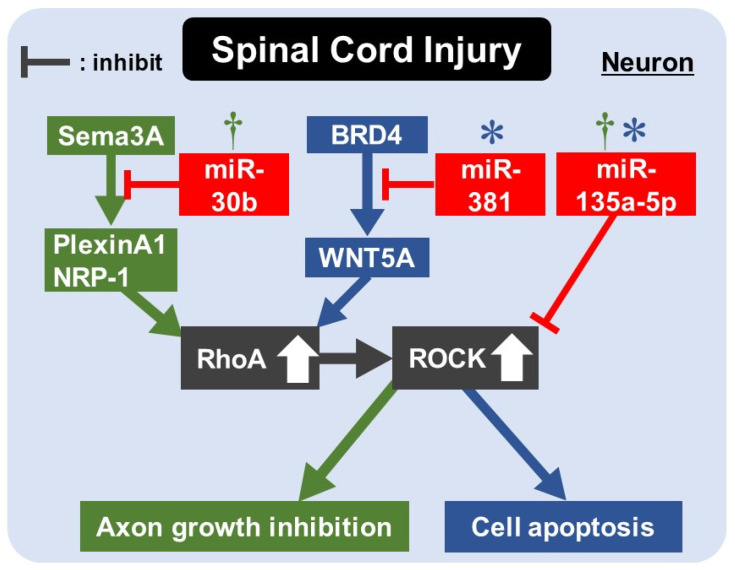
Schematic representation of the interaction between the Rho/ROCK pathway and miRNAs in spinal cord injuries. Spinal cord injuries trigger multiple signaling pathways that upregulate the Rho/ROCK pathways in neurons, leading to neuron cell apoptosis and axon growth inhibition. MiRNAs inhibit these signaling pathways, resulting in reduced cell apoptosis and the promotion of axon regeneration. † miRNA involving oxidative stress. * miRNA involving cell apoptosis. Sema3A: Semaphorin-3A; NRP-1: neuropilin-1; BRD4: bromodomain-containing protein 4; WNT5A: Wnt family member 5A; ROCK: Rho-kinase.

**Table 1 ijms-22-11573-t001:** LncRNAs/miRNAs and the Rho/ROCK pathway in ischemic strokes.

Model	lncRNA/miRNA	Expression after Insult	Target	Effects	Reference
MCAO/R in rats Hippocampal neuron of rats under OGDR	miR-190	Decreased	Rho	The overexpression of miR-190 decreased apoptosis.	[95]
MCAO/R in rats	miR-431	Decresed	Rho	The overexpression of miR-431 decreased apoptosis and promoted proliferation.	[96]
MCAO in rats PC12 cells in serum-free medium	miR-335	Decreased	ROCK2	miR-335 treatment upregulated stress granule formation, alleviated infarction, decreased ROCK2 expression, and apoptosis.	[97]
MCAO in mice N2A cells under OGD/R	miR-582-5p	Decreased	PAR-1	Overexpression of miR-582-5p inhibited the activation of the Rho/ROCK pathway by downregulating proteinase-activated receptors type-1 (PAR-1), reducing apoptosis.	[100]
MCAO/R in mice PC12 cells under OGD/R	XIST	Elevated	miR-362	XIST negatively regulated miR-362. Depletion of XIST attenuated apoptosis and inflammation via miR-362/ROCK2 axis.	[109]
	miR-362	Decreased	ROCK2		
MCAO/R in mice N2a cells under OGB/R	lncRNA-H19	Elevated	miR-148a-3p	lncRNA-H19 may act as a molecular sponge of miR-148a-3p. lncRNA-H19 altered OGD/R induced apoptosis and oxidative stress via the miR-148a-3p/ROCK2 axis.	[110]
	miR-148a-3p	Decreased	ROCK2		
MCAO/R in rats PC12 cells under OGD/R	lncRNA-SNHG14	Elevated	miR-136-5p	lncRNA-SNHG14 negatively regulated miR-136-5p as its ceRNA. lncRNA-SNHG14 promoted neurological impairment and inflammation via the miR-136-5p/ROCK1 axis.	[112]
	miR-136-5p	Decreased	ROCK1		
PC12 cells under OGD/R	lncRNA-ROR	Elevated	miR-135a-5p	lncRNA-ROR promoted oxidative damage and apoptosis via the miR-135a-5p/ROCK1/2 axis. The overexpression of miR-135a-5p decreased cell damage by inhibiting ROCK1/2.	[113]
	miR-135a-5p	Decreased	ROCK1/2		

MCAO/R: middle cerebral artery occlusion-reperfusion; OGDR: oxygen-glucose deprivation-reperfusion; ROCK: Rho-kinase; XIST: X-inactive specific transcript; SNHG14: small nucleolar RNA host gene 14; ceRNA: competing endogenous RNA; ROR: regulator of reprogramming.

## Data Availability

The data presented in this study are available on request from the corresponding author.

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
