# Peer review of "Rho/ROCK Pathway and Noncoding RNAs: Implications in Ischemic Stroke and Spinal Cord Injury"

_ijms, 2021, doi:10.3390/ijms222111573_

Round 1

Reviewer 1 Report

Ref: ijms-1425509

Title: Rho/ROCK Pathway and Noncoding RNAs: Implications in Ischemic Stroke and Spinal Cord Injury

Recommendation: Accept for publication

Very well written review article. The topic of the paper is novel and the Authors refer to the latest reports in the field. For this reason, the review will certainly find many readers. Therefore, I recommend this paper to be publish as it is.

Author Response

We want to thank Reviewer 1 for his warm comments. We are sincerely glad that reviewer 1 appreciated our review article.

Reviewer 2 Report

The manuscript provides a detailed and well-structured review on current literature regarding Rho/ROCK pathway and its' significance in ischemic stroke and spinal cord injury pathogenesis. It thoroughly describes alterations in central nervous system after ischemia and indicates the above mentioned pathway as promising therapeutic target in stroke treatment. Nevertheless, I suggest following minor revisions to increase the quality and fluidity of the manuscript:

1)  Line 38/39: "The Rho/ROCK pathway regulates a variety of critical cellular functions". 

Could you please further explain what critical cellular functions does the Rho/ROCK regulate?

2) I highly suggest to present Rho/ROCK pathway (as described in second part of the article, lines 53-66) on a graph to improve the clarity and understandability. 

3) Lines 68-70: Please describe the biogenesis of miRNAs.

4) Line 81: "IS accounts for more than 80% of all strokes, while hemorrhagic or other strokes account for the rest."

Please elaborate other types of stroke, since stroke of unknown origin constitutes an important clinical problem. 

5) Line 256-258: "Using the MCAO-reperfusion (MCAO/R) model, Jian et al. showed that Rho was a direct target of miR-190 and that overexpression of miR-190 reduced brain damage and apoptosis via the Rho/ROCK pathway."

Please elucidate how miR-190 reduced brain damage and apoptosis (inhibits the ischemia reperfusion injury and ROS formation) not only in ischemic region but also in prenumbra.

6) Line 343-345: "They found that up-regulation of miR-30b inhibited sema3A expression and RhoA/ROCK activity through PlexinA1/NRP-1 coreceptor, promoting primary sensory neuron neurite outgrowth and spinal cord sensory conductive function recovery".

I find this particularly interesting, since the downregulation of miRNA-30b expression was proposed as one of biomarkers of ischemic stroke [see article below].

Sepramaniam, S.; Tan, J.R.; Tan, K.S.; DeSilva, D.A.; Tavintharan, S.; Woon, F.P.; Wang, C.W.; Yong, F.L.; Karolina, D.S.; Kaur, P.; et al. Circulating microRNAs as biomarkers of acute stroke. Int. J. Mol. Sci. 2014, 15,1418–1432.

7) Rho-associated kinase activity was found to be an independent predictor of cardiovascular diseases, including stroke [see article below]. Please discuss in conclusions if non-coding RNAs targeting Rho/ROCK pathway (apart from being a potential therapeutic target) could also emerge as novel biomarkers of ischemic stroke / spinal cord injury.  

Kajikawa M, Noma K, Maruhashi T, et al. Rho-associated kinase activity is a predictor of cardiovascular outcomes. Hypertension. 2014;63(4):856-864. doi:10.1161/HYPERTENSIONAHA.113.02296

8) Please, double-check the manuscript for typos and grammatical mistakes. 

Reviewer 3 Report

Here I present my comments on the review entitled “Rho/ROCK Pathway and Noncoding RNAs: Implications in Ischemic Stroke and Spinal Cord Injury“ presented by Kimura et al. for publication in International Journal of Molecular Sciences (Manuscript ID: ijms-1425509)

In general, the manuscript is well written and presented in a comprehensive manner. Also figures are displayed clearly. In principle, I felt enthusiastic while I was reading the text and I enjoyed the way the authors separate the different subtopics.

As a general comment I would ask to the authors to consider to work a bit more on the redaction/ coherence of certain phrases as well as on calling a citation when it is due. Some examples of these complicated phrases or phrases lacking a direct citation are:

  • Line 82-87. Move away to the end/ or beginning of the paragraph the sentence „While severe ischemia occurs in the ischemic core, causing neuronal cell necrosis, the surrounding penumbra region cells are partially injuredwith the potential to be salvaged [24].“
  • Please add references to sentences in lines 108-100, 114-116, 118-120, 127-129, 133-136, and others.
  • Please explain better the logical link between the conceptual model NVU and IS

Also, I recommend to include the molecular work of Lisette Leyton (https://pubmed.ncbi.nlm.nih.gov/?term=Lisette+Leyton&sort=pubdate&size=20) on RhoA/ROCK in astrocytes and neurons, as it fits perfectly to the current manuscript .
